# National TB program shortages as potential factor for poor-quality TB care cascade: Healthcare workers' perspective from Beira, Mozambique

**Miguelhete Lisboa** [1,2]*, **Inês Fronteira**[2], **Paul H. Mason**[3], **Maria do Rosário O. Martins**[2]

**1** Centro de Investigação Operacional da Beira (CIOB), Instituto Nacional de Saúde (INS), Beira, Mozambique, **2** Global Health and Tropical Medicine, Instituto de Higiene e Medicina Tropical (IHMT), Universidade NOVA de Lisboa, Lisbon, Portugal, **3** School of Social Sciences, Monash University, Clayton, Australia

* miguelhetelisboa@gmail.com

## Abstract

### Background

Mozambique is one of the countries with the deadly implementation gaps in the tuberculosis (TB) care and services delivery. In-hospital delays in TB diagnosis and treatment, transmission and mortality still persist, in part, due to poor-quality of TB care cascade.

### Objective

We aimed to assess, from the healthcare workers' (HCW) perspective, factors associated with poor-quality TB care cascade and explore local sustainable suggestions to improve in-hospital TB management.

### Methods

In-depth interviews and focus group discussions were conducted with different categories of HCW. Audio-recording and written notes were taken, and content analysis was performed through atlas.ti7.

### Results

Bottlenecks within hospital TB care cascade, lack of TB staff and task shifting, centralized and limited time of TB laboratory services, and fear of healthcare workers getting infected by TB were mentioned to be the main factors associated with implementation gaps. Interviewees believe that task shifting from nurses to hospital auxiliary workers, and from higher and well-trained to lower HCW are accepted and feasible. The expansion and use of molecular TB diagnostic tools are seen by the interviewees as a proper way to fight effectively against both sensitive and MDR TB. Ensuring provision of N95 respiratory masks is believed to be an essential requirement for effective engagement of the HCW on high-quality in-hospital

**Data Availability Statement:** The dataset generated and analysed during the current study is not publicly available due to confidentiality reasons although an anonymized minimal dataset could be

provided to any researcher upon reasonable formal request to the Internal Scientific Committee of Centro de Investigação Operacional da Beira, Instituto Nacional de Saúde, at geral@ciob.gov.mz or Rua Correia Brito, #1323 – Pontagea, Beira, Moçambique.

**Funding:** This study was elaborated based on the work of first Miguelhete Lisboa's doctoral program, a Fundação Calouste Gulbenkian (FCG) scholarship holder (ID: P-135647/SBG/2014) and, used grants obtained from World Health Organization, The Special Programme for Research and Training in Tropical Diseases (WHO/TDR) and co-sponsored by the United Nations Children's Fund, United Nations Development Programme, World Bank and WHO – award ID number: B40151/2014. The FCG and WHO/TDR were neither involved in the design of the study and collection, analysis, interpretation of data, nor in the writing of manuscript or decision to publish. Therefore, the authors are responsible for all information.

**Competing interests:** The authors have declared that no competing interests exist.

TB care. For monitoring and evaluation, TB quality improvement teams in each health facility are considered to be an added value.

## Conclusion

Shortage of resources within the national TB control programme is one of the potential factors for poor-quality of the TB care cascade. Task shifting of TB care and services delivery, decentralization of the molecular TB diagnostic tools, and regular provision of N95 respiratory masks should contribute not just to reduce the impact of resource scarceness, but also to ensure proper TB diagnosis and treatment to both sensitive and MDR TB.

## Introduction

One of the main tuberculosis (TB) reduction strategies is the early detection and rapid administration of proper anti-TB treatment [1–2]. Studies in multiple African countries have shown that diagnostic and treatment delays persist, in part, due to healthcare system or in-ward delays related to the shortage of resources, particularly the well-trained healthcare workers (HCW) and molecular diagnostic tools within the peripheral health facilities [3–4].

The implementation of the national policies on TB infection control (TBIC) measures in healthcare facilities is seen to be a specific matter for healthcare workers, particularly for the nurses [5], however, nurses are overwhelmed in Mozambique, as the country´s nurse to population ratio is about 2.9 per 10 000 inhabitants. In addition, there is still shortage of medical doctors and well-trained laboratory technicians (superior level), as the population ratio is about 0.5 and 0.3 per 10 000 inhabitants, respectively [6].

In-hospital delays of TB diagnosis and treatment, transmission and mortality persist around the country, in part, due to poor-quality of TB care cascade [7–9]. Shortage of resources seems to hinder the local authority commitment on reduction of delays in TB diagnosis and treatment, morbidity and mortality [8–10].

Task shifting to tackle health worker shortages in the context of the HIV epidemic, aiming to strengthen and expand the health workforce was endorsed by world health organization [11] and is a well-known, cost-effective strategy, also approved and recommended by Mozambique ministry of health [12]. However, little is known about the task shifting of TB services delivery to tackle the human resource crisis in the face of the TB and emerging epidemic of multidrug resistant (MDR) TB.

Investiment in molecular TB diagnostic tools to the district level is still delayed, as most of health facilities are still using acid-fast bacilli smear microscopy as their first tool for TB diagnosis, despite the emerging epidemic of MDR TB. TB care and services (laboratories and treatment centres) are only available 8 hours per day and none at all, during weekend and holidays [8]. In addition, within the Mozambique Ministry of Health, there is not any reliable transport network for clinical specimens, including sputum samples, from the peripheral health facilities to the TB laboratory with molecular diagnostic tools, most of them located at district or provincial levels.

This study was carried out to assess, from the healthcare workers' perspective, the factors associated with in-hospital poor-quality TB care cascade and explore local sustainable suggestions to improve in-hospital TB care services.

## Material and methods

### Study design, population and sample

This was a cross-sectional qualitative study [13] conducted from January to February 2019.

The target population was the health staff from the Beira Central Hospital dealing with TB matters. Therefore, there were a total of 71 healthcare workers including managers and decision-makers. Among them, there were 11 medical doctors, 16 nurses, 13 laboratory technicians and 31 hospital auxiliary workers. We used a convenience sampling approach to select our sample [13]. The selected categories of health workers represent the full range of healthcare staff with different levels of training and experience in relation to TB. They also had different responsibilities and possibly had different perceptions and practices related to TB diagnosis, treatment and infection control measures within the central hospital.

Our sample was purposively only made of staff working in the medical wards and TB laboratory, hence, 20 hospital auxiliary workers, 10 nurses, 5 medical doctors, 9 TB laboratory technicians. Additionally, 7 managers and 3 hospital decision-makers were also selected and asked to participate in the focus group discussions (FGD) and individual in-depth interviews (IDI).

### Study setting, TB diagnosis and treatment capabilities

The study was carried out in Beira Central Hospital. Beira city (the capital of Sofala province) is located in the central region of Mozambique (about 1,200 km northern of Maputo, the capital city of Mozambique). Beira is the second largest city in Mozambique with a population estimated in about 463,442 inhabitants in 2017. The TB incidence rate (659/100,000 inhabitants) and TB/HIV co-infection rate (about 63%) in Beira city are above the national average [14].

The Beira Central Hospital, a quaternary level health facility, is a referral facility for 3 provinces and partly for Northern region of Inhambane province. This hospital is equipped with high skilled personal (Infectious Disease Doctor, Pulmonologists, Laboratory Technicians, Imagiologist, etc.) and TB laboratory technology, including GeneXpert MTB/Rifampicin, culture and drug sensitivity testing in TB and, where most of TB suspected patients from these provinces are referred to and diagnosed. The Beira Central Hospital has 4 medical wards with approximately 250 beds and 1 separate TB ward with about 34 beds [14].

### Data collection and procedures

FGD were used to assess a broad range of views about TB within medical wards from auxiliary workers, nurses and TB laboratory technicians. IDI were used to address individual experiences or feelings from those very busy staff like medical doctors, managers and hospital decision-makers. The FGD and IDI guides were previously discussed with Infectious Disease Doctor, TB and Laboratory Experts, piloted and approved for its use.

There were two FGD with auxiliary workers (10 females and 10 males), one with nurses (5 females and 5 males), and one with TB laboratory technicians (4 femeles and 5 males). We conducted fifteen IDI with five medical doctors, seven managers and three hospital decision-makers. All FGD and IDI were carried out in Portuguese.

The FGD and IDI were held in the meeting room and individual offices respectively, depending on the availability of the health-care workers. During the FGD, one moderator led each discussion using a standard guide and two investigators were observing and taking notes from the discussion. For IDI, one moderator led the sessions and one investigator observed and took notes during the interviews. The moderator and observers were the same in all FGD

and IDI. Each FGD and IDI was conducted in an average time of 80 and 60 minutes, respectively.

In undertaking the FGD and IDI sessions digital recordings were done. A guide was used containing the following areas: general situation of nosocomial TB, in-hospital TB care cascade gaps, probable factors associated with in-hospital delayed TB diagnosis and treatment, acceptability of auxiliary workers and 24-hour TB services and, what should be done to ensure feasibility of these last two approaches.

After finishing each FGD, the study team discussed the points as a way of preparing for the forthcoming discussions. Data were processed on a daily basis, immediately after the FGD or IDI. There was no need to change the FGD and IDI guide.

## Data analysis

Discussions and interviews were transcribed verbatim. After transcription, content analysis using inductive analysis [13] initially consisted of the identification of transcripts that were linked to the objectives of the study, preparation of analysis codes, transcript text encoding and analysis between the codes. Then, the study team again discussed all codes and agreed the final code for the encoding and the final analysis. Code group in comprehensive themes representing the contents of the objectives (the factors associated with in-hospital poor-quality TB care cascade and explore local sustainable suggestions to improve in-hospital TB care services) was conducted subsequently.

Finally, relationships between topics using the technique of examination and re-examination of the relevant parts were established. Quotes that represented the emerging themes were selected for inclusion in the manuscript.

Atlas-ti version 7 for Windows was used to facilitate the search, classification and organization of the data.

## Ethical statement

The study protocol was approved by the Institutional Bioethics Committee for Health of the National Institute of Health of Mozambique and by WHO Ethics Review Committee. The Mozambique Ministry of Health, the Sofala Provincial Health and Beira Central Hospital authorities provided administrative approval. After an introduction of the participants, researchers and the objectives of the study, written informed consent procedures were ensured. Therefore, all participants signed an informed consent form. Participants were allowed to decline to take part of the FGD or IDI. To ensure that every healthcare worker had the opportunity to contribute freely in the FGD, the groups were homogeneous in terms of responsibilities.

The anonymity of the participants was guaranteed by their identification with the initial letters of their professional category followed by the numeral, according to the chronological order of the interviews or sitting position during the focus discussion groups, for example, MD1 (Medical Doctor 1), DM3 (Decision-Maker 3), AW5 (Auxiliary Worker 5), M3 (Manager 3), LT9 (Laboratory Technicians 9), and so on.

## Results

### Sociodemographic attributes of the respondents

A total 54 key informants were purposively selected and volunteered to participate. Fifteen participated in IDI and 39 in FGD. From the total, 52% (n = 54) were male, the mean (±SD)

**Table 1. Sociodemographic attributes of the respondents from Beira, Mozambique, 2019.**

| Characteristics | Number | Percentage (%) |
|---|---|---|
| **Sex** | | |
| Male | 28 | 51.9 |
| Female | 26 | 48.1 |
| **Age groups** | | |
| 24–35 years old | 33 | 61.1 |
| 36–45 years old | 17 | 31.5 |
| 46–55 years old | 4 | 7.4 |
| **Profession / responsability** | | |
| Physician (decision-maker) | 3 | 5.6 |
| Physician (ward assistant) | 5 | 9.3 |
| Nurse (ward officer) | 5 | 9.3 |
| Nurse (ward assistant) | 10 | 18.5 |
| Laboratory technician (officer) | 2 | 3.7 |
| Laboratory technician (assistant) | 9 | 16.7 |
| Auxiliary worker | 20 | 37.0 |
| **Professional experience on TB matters (years)** | | |
| 1–3 years | 17 | 31.5 |
| 4–6 years | 24 | 44.4 |
| ≥ 7 years | 13 | 24.1 |

age was 34 (±7) years and 68.5% of the participants had more than four years working experience in TB matters—Table 1.

## Main themes emerged

The agreed emerged themes were in-hospital TB care cascade gaps, lack of TB staff and task shifting, centralized and limited time of TB laboratory services and fear of healthcare workers getting infected by TB.

## In-hospital TB care cascade gaps

From suspected TB patient admission to the treatment initiation, according to interviewees, there are four mandatory steps. These steps and its related factors are possible TB care cascade gaps that may contribute to increased in-hospital delays to TB diagnosis and treatment, onward TB transmission and TB mortality in the Beira Central Hospital–Fig 1.

Additionally, the TB care cascade is getting worse as there is a lack of motivated TB taskforce responsible for organizing and coordinating regular discussion of TB data, supervision, monitoring and evaluation of the TB activities within the health facilities.

## Lack of staff and task shifting

Interviewees were first asked to identify which category of the health workers should be considered for task shifting of TB care provision and what needs to be done. Almost all respondents (n = 51) indicated that staff shortage, particularly of nurses, is negatively impacting the quality of the TB care delivery. Therefore, as opposed to an in-ward nurse, in-hospital expedition of TB related issues could best be acceptable if hospital auxiliary workers are carefully selected, trained and assigned to logistic and operational issues in relation to sputum collection from medical ward to TB laboratory, and examination results collection from TB laboratory to

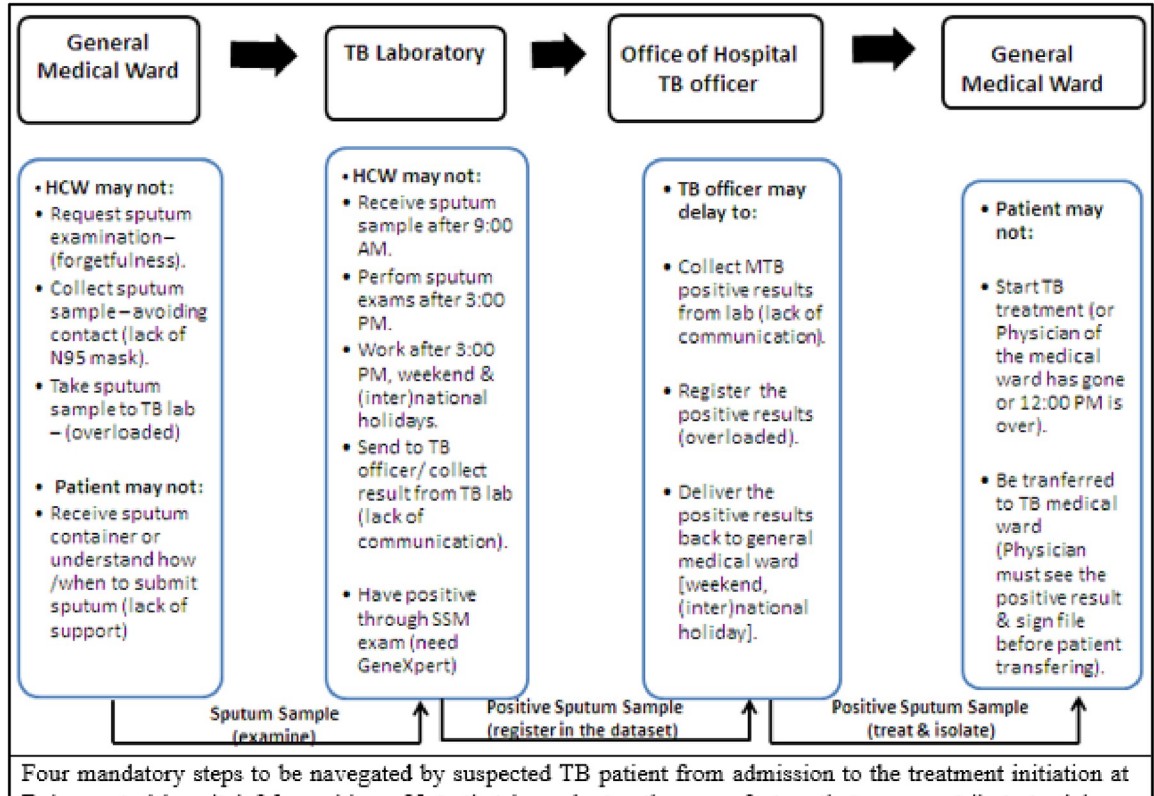

**Fig 1. Current steps and perceived factors associated with in-hospital delays to TB diagnostic and treatment in Beira, Mozambique, 2019.**

medical doctors or any other trained staff for early TB treatment initiation, TB patient isolation and education.

The task shifting was also suggested from higher-trained professionals (ID doctors, Pneumologist doctors, etc) to lower-trained workers (technicians of medicine, laboratory, nursing, etc) specially those working at peripheral health facilities.

As five out of seven managers during a focus group also supported engagement and task shifting from nurses to hospital auxiliary workers. This is illustrated by the speech of one of the interviewees—a manager during a focus group discussion:

*Auxiliary workers have always been involved in sputum sample collection and sending to the TB laboratory (. . .), however, they are not seriously assigned to this task and they don´t feel as their own responsibility (. . .). What should be done is to select, train, engage on TB matters as expediter and supervise them. To ensure supervion, every health facility should have a high-quality TB care committee and regular coordination/discussion and evaluation meetings (. . .)*

(M2).

In addition, all medical doctors, during an interview, have supported the need of training the TB front line staff, particularly at peripheral health facilities and district levels, on proper management of multidrug resistant TB. One of the medical doctors stated:

*(. . .) TB cases are everywherewe and effective management is lacking due to the shortage of well-trained health workers on TB matters, especially when multidrug resistant (MDR) TB is suspected/diagnosed at district or peripheral levels (. . .), task shifting from medical doctors to lower health workers for high-quality TB care at rural areas is urgent to overcome the challenges of the MDR TB management*

(MD3).

## Centralized and limited time of TB laboratory services

Due to shortage of laboratory technicians there is a lack of availability of TB services compared to other clinical services, which was emphasised as an important factor affecting negatively the early diagnostic and treatment of TB in Beira. Thus, the respondents believe that the lack of 24-hour TB laboratory services provision seems to contribute to underutilisation of the scarceness molecular diagnostic tools and lately, on delay in TB diagnosis and treatment initiation.

Importantly, beside the resource constrainst, the current centralized molecular diagnostic tools (e.g. GeneXpert) approach was mentioned to be one of the most potential shortcomings, as the first door of entrance of the most of patients is at peripheral level, within the primary healthcare, decentalization of the molecular diagnostic tools are seen to be urgent.

This finding shortfall was raised as a concern by almost all participants (n = 38) during all focus groups and in most of the interviews (n = 12) as illustrated by the following interviews of a physician:

*As a national routine, during working-days, TB laboratories are open from 7:30 AM, and close at 3:30 PM, but sputum sample delivery to the TB laboratory ends at 9:00 AM every day. Additionally, sputum sample examination is not performed on weekend or (inter)national holidays. How can we reduce nosocomial TB or in-hospital TB mortality?*

(MD4).

All laboratory technicians during a focus group also emphasized the following after a laboratory techinician said:

*We are happy to guarantee 24-hour TB laboratory services, but it would be necessary to install a micro-TB laboratory in the emergency department and train the lower laboratory technician to use the rapid molecular diagnostic tools (Xpert). (. . .) working on shift/schedule basis, we should overcome the period of time that TB laboratory is closed and Xpert is underused*

(LT9).

In addition, a decision-maker during an interview stated:

*(. . .) ideally, every districts and/or peripheral health facilities should be equipped with Xpert machine and its 24-hours availability for early and proper TB diagnosis and treatment. Old TB diagnostic tools (microscopes) are not helping anymore as most of TB cases are HIV co-infected or are drug resistant TB, which is not diagnosed through acid-fast bacilli smear microscopy*

(DM2).

## Fear of healthcare workers getting infected by TB

Interviewees said that the poor-quality of TB care delivery and associated in-hospital TB delays seem to be also due to fear of healthcare workers getting infected by TB, poor motivation, lack of supervision to perform their normal duties on TB services swiftly. Frequent stockout of N95 respiratory masks and irregular provision of them to the healthcare workers, particularly for the front-line TB staff, is believed to be one of the causes of in-hospital TB delays at Beira central hospital. Therefore, healthcare workers may neglect to be in contact or managing TB patient due to fear of getting infected by TB.

A general practicioner during an interview stated:

*(. . .) Very importantly, regular provision of N95 masks is an essential condition to motivate HCW to be in closer contact with the TB patients and to work willingly, overcoming the fear of contamination (. . .)*

(MD2).

On the other hand, an auxiliary worker (with strong support from other auxiliary workers) during a focus group discussion said:

*(. . .) we have learnt from several trainings and now not looking after in good manner the TB patients as just we´re protecting ourselves. . . (. . .) we are ready to keep much more attention and seriously the TB logistic and operational issues, if the hospital directorate provides the N95 masks and gloves for our own protection*

(HAW7).

## Discussion

The study results suggested that task shifting of in-hospital TB services delivery from nurses to hospital auxiliary workers, especially for expedition of the TB diagnosis and treatment matters, is acceptable and feasible approach. Interviewees believed that hospital auxiliary workers should sustainably ensure that sputum sample is promptly collected and delivered to TB laboratories, smear microscopy and/or Xpert assay results are collected from TB laboratories and delivered to physicians, proper treatment is started immediately and, infectious pulmonary TB patients are isolated as soon as the diagnosis is made. In addition, task shifting was mentioned from the TB management perspective, meaning to higher professionals to lower healthcare workers.

This result is completely a new finding as the implementation of the TB infection control (TBIC) measures in healthcare facilities is seen to be a specific matter for healthcare workers, particularly for nurses [5]. Unfortunately, the nurses and medical doctors in Mozambique are overwhelmed [6]. Therefore, considering task shifting from nurses to hospital auxiliary workers on the logistic and operationalization of TB diagnosis and treatment, and to many lower technicians (laboratory, medicine, nurses) for proper TB management at peripheral level, should be one of the strategies to strenghthen the in-hospital quality of TB care cascade. In addition, having a high-quality TB service delivery committee in every health facility, to train, supervise, monitor and evaluate the process of task shifting should be an addition value to be considered.

The study results also demonstrated that interviewees are aware of the inefficiency of acid-fast bacilli smear microscopy as the first-choice diagnostic tool, and the centralized and limited

time of the TB laboratory services. Therefore, the interviewees believe that having a TB laboratory with molecular TB diagnostic tool within the emergency department but also decentralizing the molecular diagnostic tools, task shifting from higher to lower laboratory technicians, and ensuring 24-hours work-shift, should reduce the current TB care cascade gaps and associated in-hospital TB delays due to TB laboratory operational delays.

This finding is in line with the streamlined TB diagnosis and treatment initiation strategy implemented in the peripheal health facilities in Uganda [15], the implementation of GenoType MTBDRplus at Brewelskloof hospital in South Africa [16], and the implementation of refocused, intensified, administrative tuberculosis transmission control strategy using molecular diagnostic tools in TB hospitals in Russia and Bangladesh [5], not only to speed up the diagnosis and treatment of both, sensitive to and MDR TB, but also to strenghten the implementation of the TBIC measures within health facilities toward to zero nosocomial TB transmission.

From the decentralization of molecular diagnostic tools (from provincial to peripheral health facilities level) point of view, has been analysed and considered to be cost-effective and likely to ensure high-quality TB care cascade, particularly in settings where TB morbimortality due to TB delays, and rates of lost to follow are high, [17] like Mozambique.

One of the study results was also the clear demonstration of assumed negligence from all categories of the healthcare workers on proper management of TB patients due to fear of being infected resulting from a lack of provision of N95 respiratory masks, despite of TB infection prevention and control trainings received.

These findings are similar to those described by Brouwer et al [18] and Engelbrecht et al [19] in terms of the relationship between shortage of the N95 respiratory masks and lack of provision of compassionate and high-quality TB care from the healthcare workers.

It suggests that training only healthcare workers and having guidelines is not enough to have the TB infection prevention and control measures under proper implementation. Therefore, if TB care cascade is to be improved within the health facilities, national TB program should also prioritize in solving the shortages within the TB care cascade, particularly ensuring availability of personal respiratory protection.

This study has several limitations: first, the external validity might be affected by its cross-sectional analysis, relatively small sample size and its conduction only in health facility of quaternary level and; second, qualitative analysis approach may have resulted in sections of data being misinterpreted due to thematic aggregation or due to the tendency for certain types of socially acceptable opinion to emerge, and for certain types of participant to dominate the research process, despite measures were taken by separating socio-professional categories, selection and involvement of skilled moderators. However, the results of this study may be applied to similar settings.

## Conclusion

Programatic manageable shortages within the national TB program still remain as one of the potential factors for poor-quality of the TB care cascade. Therefore, strenghthening the national TB program should be prioritized. Task shifting of in-hospital TB services delivery from nurses to hospital auxiliary workers, and to lower many technicians are accepted and feasible. Ensuring 24-hours molecular TB diagnostic tools or at least decentralizing them should contribute not just to reduce TB delays, but also to ensure proper TB treatment to both sensitive and MDR TB. In addition, provision of N95 respiratory masks is believed to be an essential requirement for effective engagement of the healthcare workers on high-quality in-hospital TB care, alongside TB quality improvement teams in each health facility to ensure an effective implementation, monitoring and evaluation.

## Supporting information

**S1 File. Prompts used in the individual in-depth interview and focus group discussions.**
Supporting information of manuscript "*National TB program shortages as potential factor for poor-quality TB care cascade*: *healthcare workers' perspective from Beira, Mozambique*".
(DOCX)

## Acknowledgments

The authors acknowledge the Tutorial Commission of the doctoral program of Miguelhete Lisboa (Professors Sonia Dias and Miguel Viveiros at Instituto de Higiene e Medicina Tropical Universidade Nova de Lisboa, Portugal); people who helped in the collection and data management: Marques Nhamonga, Joaquim Lequechane and Estefano Colove; the Centro de Investigação Operacional da Beira directorate and all colleagues, the Beira Central Hospital directorate and all healthcare workers.

## Author Contributions

**Conceptualization:** Miguelhete Lisboa, Paul H. Mason.

**Data curation:** Miguelhete Lisboa.

**Formal analysis:** Miguelhete Lisboa, Inês Fronteira, Paul H. Mason, Maria do Rosário O. Martins.

**Funding acquisition:** Miguelhete Lisboa.

**Investigation:** Miguelhete Lisboa.

**Methodology:** Miguelhete Lisboa, Inês Fronteira, Paul H. Mason, Maria do Rosário O. Martins.

**Project administration:** Miguelhete Lisboa.

**Resources:** Miguelhete Lisboa.

**Software:** Miguelhete Lisboa.

**Supervision:** Miguelhete Lisboa, Paul H. Mason.

**Validation:** Miguelhete Lisboa, Paul H. Mason.

**Visualization:** Miguelhete Lisboa, Paul H. Mason.

**Writing – original draft:** Miguelhete Lisboa, Inês Fronteira, Paul H. Mason, Maria do Rosário O. Martins.

**Writing – review & editing:** Miguelhete Lisboa, Inês Fronteira, Paul H. Mason, Maria do Rosário O. Martins.

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
