## [Decision Letter · Decision Letter 0]

26 Nov 2019

PONE-D-19-22633

National TB program shortages as potential factor for poor-quality TB care cascade: healthcare workers’ perspective from Beira, Mozambique

PLOS ONE

Dear Dr. Lisboa,

Thank you for submitting your manuscript to PLOS ONE. After careful consideration, we feel that it has merit but does not fully meet PLOS ONE’s publication criteria as it currently stands. Therefore, we invite you to submit a revised version of the manuscript that addresses the points raised during the review process.

We would appreciate receiving your revised manuscript. To enhance the reproducibility of your results, we recommend that if applicable you deposit your laboratory protocols in protocols.io, where a protocol can be assigned its own identifier (DOI) such that it can be cited independently in the future. For instructions see: http://journals.plos.org/plosone/s/submission-guidelines#loc-laboratory-protocols

We look forward to receiving your revised manuscript.

Kind regards,

Frederick Quinn

Academic Editor

PLOS ONE

Journal Requirements:

1. Please provide additional details regarding participant consent. In the ethics statement in the Methods and online submission information, please ensure that you have specified (1) whether consent was informed and (2) what type you obtained (for instance, written or verbal, and if verbal, how it was documented and witnessed). If your study included minors, state whether you obtained consent from parents or guardians. If the need for consent was waived by the ethics committee, please include this information.

Reviewers' comments:

Reviewer's Responses to Questions

**Comments to the Author**

1. Is the manuscript technically sound, and do the data support the conclusions?

Reviewer #1: Yes

Reviewer #2: Yes

2. Has the statistical analysis been performed appropriately and rigorously? 

Reviewer #1: N/A

Reviewer #2: N/A

3. Have the authors made all data underlying the findings in their manuscript fully available?

Reviewer #1: No

Reviewer #2: Yes

4. Is the manuscript presented in an intelligible fashion and written in standard English?

Reviewer #1: Yes

Reviewer #2: No

5. Review Comments to the Author

Reviewer #1: Authors should address these minor comments

ABSTRACT:

(a) In the background it is not very clear what you intend to communicate. Please rephrase for clarity.

(b) The last part of the "Objective) i.e. "---to overcome them toward to the TB elimination" should be revisited.

(c) Line 67 and throughout the manuscript: Write "lab" in full i.e. "laboratory"

INTRODUCTION:

Lines 121-123 should be rewritten for clarity.

MATERIALS and METHODS:

(a) Line 127: Should be "This was a cross-sectional qualitative study(13) conducted from January to February 2019".

(b) Lines 135-138: Please provide the total number of health workers in the Beira Central Hospital against the indicated health categories i.e. nurses, medical doctors etc.

(c) Lines 141-142: For an average reader, please describe where Beira city is located within Mozambique i.e. northern, central, eastern etc.

(d) Line 152: Should be "FGD were used to assess a broad range of views about --------"

(e) Line 159-160: Should be "We conducted fifteen IDI with five medical doctors, seven managers and three hospital decision-makers"

(f) Lines 161-164: Although the consent procedure is mentioned, please mention whether the study obtained ethics approval from an ethics committee?

RESULTS:

(a) Line 265: Correct the typo i.e. should be "resource"

(b) Line 292: Fear of contamination is frequently mentioned in the text but what do you mean exactly? Do you mean contamination of TB cultures or health workers getting infected by TB (as the context of lines 292-298 suggest?). Clarify.

Reviewer #2: The authors target the programmatic manageable shortages, specifically the task shifting inefficiency of TB services, that persist in regions with poor quality of TB management. The authors seek to understand the root causes of poor implementation of task shifting strategy and draw a conclusion from narrative survey-based data. The overall logic is sound, scientifically. However, given that only healthcare workers, who have been implementing guided TB services and strategies, were included the in this study, how do you justify the insight of the interviewees? Or is there any evidence suggesting the identified factors extracted from the interviews are the ones playing the most crucial role in the quality of TB services?

In addition, please monitor the writing throughout the manuscript. There are typos and confusing sentences in the manuscript. For example, line 55, line 77-78, line 101-104, Table 1 Percentage "51, 9" or "51.9", line 319.

6. PLOS authors have the option to publish the peer review history of their article (what does this mean?). If published, this will include your full peer review and any attached files.

Reviewer #1: No

Reviewer #2: No

---

## [Author Response · Author response to Decision Letter 0]

8 Jan 2020

Reviewers’ comments and authors’ responses to each point raised by the reviewers

Responses to reviewer #1

ABSTRACT:

(a) In the background it is not very clear what you intend to communicate. Please rephrase for clarity.

The content of the background (under abstract section) was revised and now is written as following: Mozambique is one of the countries with the deadly implementation gaps in the tuberculosis (TB) care and services delivery. In-hospital delays in TB diagnosis and treatment, transmission and mortality still persist, in part, due to poor-quality of TB care cascade.

(b) The last part of the "Objective) i.e. "---to overcome them toward to the TB elimination" should be revisited.

The content of the objective (under abstract section) was rewritten as following:

We aimed to assess, from the healthcare workers’ (HCW) perspective, factors associated with poor-quality TB care cascade and explore local sustainable suggestions to improve in-hospital TB care and services delivery.

(c) Line 67 and throughout the manuscript: Write "lab" in full i.e. "laboratory"

Thanks for the excellent comment. The word was revised on every single page of the document and now is written in full “laboratory” instead of lab

INTRODUCTION:

Lines 121-123 should be rewritten for clarity.

The content of the lines (under introduction section) were rewritten as following:

This study was carried out to assess, from the healthcare workers’ perspective, the factors associated with in-hospital poor-quality TB care cascade and explore local sustainable suggestions to improve in-hospital TB care services.

MATERIALS and METHODS:

(a) Line 127: Should be "This was a cross-sectional qualitative study (13) conducted from January to February 2019".

This sentence was corrected as recommended. “This was a cross-sectional qualitative study (13) conducted from January to February 2019”

(b) Lines 135-138: Please provide the total number of health workers in the Beira Central Hospital against the indicated health categories i.e. nurses, medical doctors etc.

Under the study design, population and sample section, is rewritten as:

The target population was the health staff from the Beira Central Hospital dealing with TB matters. Therefore, there were a total of 71 healthcare workers including managers and decision-makers. Among them, there were 11 medical doctors, 16 nurses, 13 laboratory technicians and 31 hospital auxiliary workers. We used a convenience sampling approach to select our sample(13). The selected categories of health workers represent the full range of healthcare staff with different levels of training and experience in relation to TB. They also had different responsibilities and possibly had different perceptions and practices related to TB diagnosis, treatment and infection control measures within the central hospital. 

Our sample was purposively only made of staff working in the medical wards and TB laboratory, hence, 20 hospital auxiliary workers, 10 nurses, 5 medical doctors, 9 TB laboratory technicians. Additionally, 7 managers and 3 hospital decision-makers were also selected and asked to participate in the focus group discussions (FGD) and individual in-depth interviews (IDI).

(c) Lines 141-142: For an average reader, please describe where Beira city is located within Mozambique i.e. northern, central, eastern etc.

Under the study setting section, the paragraph was rewritten as:

The study was carried out in Hospital Central da Beira. Beira city (the capital of Sofala province) is located in the central region of Mozambique (about 1,200 km northern of Maputo, the capital city of Mozambique). Beira is the second largest city in Mozambique with a population estimated in about 463,442 inhabitants in 2017. The TB incidence rate (659/100,000 inhabitants) and TB/HIV co-infection rate (about 63%) in Beira city are above the national average(14). 

(d) Line 152: Should be "FGD were used to assess a broad range of views about -----"

This sentence was corrected as recommended. “FGD were used to assess a broad range of views about TB within medical wards from auxiliary workers, nurses and TB laboratory technicians”

(e) Line 159-160: Should be "We conducted fifteen IDI with five medical doctors, seven managers and three hospital decision-makers"

This sentence was corrected as recommended. “We conducted fifteen IDI with five medical doctors, seven managers and three hospital decision-makers”

(f) Lines 161-164: Although the consent procedure is mentioned, please mention whether the study obtained ethics approval from an ethics committee?

The authors have included an ethical statement (under “methods and material” section – on page 08) as following:

“The study protocol was approved by the Institutional Bioethics Committee for Health of the National Institute of Health of Mozambique and by WHO Ethics Review Committee. The Mozambique Ministry of Health, the Sofala Provincial Health and Beira Central Hospital authorities provided administrative approval. After an introduction of the participants, researchers and the objectives of the study, written informed consent procedures were ensured. Therefore, all participants signed an informed consent form. Participants were allowed to decline to take part of the FGD or IDI. To ensure that every healthcare worker had the opportunity to contribute freely in the FGD, the groups were homogeneous in terms of responsibilities. The anonymity of the participants was guaranteed by their identification with the initial letters of their professional category followed by the numeral, according to the chronological order of the interviews or sitting position during the focus discussion groups, for example, MD1 (Medical Doctor 1), DM3 (Decision-Maker 3), AW5 (Auxiliary Worker 5), M3 (Manager 3), LT9 (Laboratory Technicians 9), and so on.”

RESULTS:

(a) Line 265: Correct the typo i.e. should be "resource"

The typo was corrected. Now is written “resource”

(b) Line 292: Fear of contamination is frequently mentioned in the text but what do you mean exactly? Do you mean contamination of TB cultures or health workers getting infected by TB (as the context of lines 292-298 suggest?). Clarify.

Thanks for the excellent comment. The sentence was revised on every single page of the document and now is clear that we’re referring to: “fear of healthcare workers getting infected by TB”

Responses to reviewer #2

The authors target the programmatic manageable shortages, specifically the task shifting inefficiency of TB services, that persist in regions with poor quality of TB management. The authors seek to understand the root causes of poor implementation of task shifting strategy and draw a conclusion from narrative survey-based data. The overall logic is sound, scientifically. However, given that only healthcare workers, who have been implementing guided TB services and strategies, were included the in this study, how do you justify the insight of the interviewees? Or is there any evidence suggesting the identified factors extracted from the interviews are the ones playing the most crucial role in the quality of TB services?

Thank you very much for this comment. 

As we stated in the paper, the aim of this study was to understand, from the healthcare workers’ perspective, the cause of causes of poor quality of TB care cascade within the hospital environment. As described by Patton, M. (1990) purposeful typical case sampling - interviewing key-healthcare cadres who are dealing with, are knowledgeable and have experience on TB care and services delivery and management - can help identify what are and not typical concerns, rather than general healthcare workers or patients. 

Therefore, we do believe that the factors extracted from the interviews are the ones playing the most crucial role in the quality of TB services, at least at the Beira Central Hospital. However, we are keeping in mind that the purpose of typical cases is to describe and illustrate what is and not typical to those unfamiliar with the program, and not to make generalized statements about the experiences of all participants.

In addition, please monitor the writing throughout the manuscript. There are typos and confusing sentences in the manuscript. For example:

line 55, - the background of the abstract was revised and rewritten as:

In-hospital tuberculosis (TB) delays, transmission and mortality persists in Mozambique, in part, due to poor-quality of TB care cascade. Little is known about the local factors associated with the deadly implementation gaps in the TB care and services delivery.

line 77-78, - the conclusion of the abstract was revised and was rewritten as:

Shortage of resources within the national TB control programme is one of the potential factors for poor-quality of the TB care cascade. Task shifting of TB care and services delivery to lower HCW, decentralization of the molecular TB diagnostic tools, and regular provision of N95 respiratory masks should contribute not just to reduce the impact of resource scarceness, but also to ensure proper TB diagnosis and treatment to both sensitive and MDR TB. 

line 101-104, - the content of the lines (under introduction section) was revised and rewritten as:

In-hospital delays of TB diagnosis and treatment, transmission and mortality persist around the country, in part, due to poor-quality of TB care cascade (7-9). Shortage of resources seems to hinder the local authority commitment on reduction of delays in TB diagnosis and treatment, morbidity and mortality (8-10). 

Table 1 Percentage "51, 9" or "51.9", 

The table content and typos were revised and can be found on the page 9 of the doc in track changes.

line 319. - the content of the sentence (under discussion section) was revised and rewritten:

In addition, task shifting was mentioned from the TB management perspective, meaning to higher professionals to lower healthcare workers.

---

## [Decision Letter · Decision Letter 1]

28 Jan 2020

National TB program shortages as potential factor for poor-quality TB care cascade: healthcare workers’ perspective from Beira, Mozambique

PONE-D-19-22633R1

Dear Dr. Lisboa,

We are pleased to inform you that your manuscript has been judged scientifically suitable for publication and will be formally accepted for publication once it complies with all outstanding technical requirements.

With kind regards,

Frederick Quinn

Academic Editor

PLOS ONE

Additional Editor Comments (optional):

Reviewers' comments:

Reviewer's Responses to Questions

**Comments to the Author**

1. If the authors have adequately addressed your comments raised in a previous round of review and you feel that this manuscript is now acceptable for publication, you may indicate that here to bypass the “Comments to the Author” section, enter your conflict of interest statement in the “Confidential to Editor” section, and submit your "Accept" recommendation.

Reviewer #1: All comments have been addressed

Reviewer #2: All comments have been addressed

2. Is the manuscript technically sound, and do the data support the conclusions?

Reviewer #1: Yes

Reviewer #2: Yes

3. Has the statistical analysis been performed appropriately and rigorously? 

Reviewer #1: N/A

Reviewer #2: N/A

4. Have the authors made all data underlying the findings in their manuscript fully available?

Reviewer #1: Yes

Reviewer #2: Yes

5. Is the manuscript presented in an intelligible fashion and written in standard English?

Reviewer #1: Yes

Reviewer #2: Yes

6. Review Comments to the Author

Reviewer #1: Authors have satisfactorily responded to all my comments. I recommend publication of this manuscript without further revisions.

Reviewer #2: The authors have addressed all the comments properly. The structure of the manuscript, the introduction of the study goal, the data presentation, the insight have been polished to a publication quality.

7. PLOS authors have the option to publish the peer review history of their article (what does this mean?). If published, this will include your full peer review and any attached files.

Reviewer #1: Yes: David Patrick Kateete

Reviewer #2: No

---

## [Editor Report · Acceptance letter]

30 Jan 2020

PONE-D-19-22633R1 

National TB program shortages as potential factor for poor-quality TB care cascade: healthcare workers’ perspective from Beira, Mozambique 

Dear Dr. Lisboa:

I am pleased to inform you that your manuscript has been deemed suitable for publication in PLOS ONE. Congratulations! Your manuscript is now with our production department. 

With kind regards,

on behalf of

Dr. Frederick Quinn 

Academic Editor

PLOS ONE